# Carborane-Containing Folic Acid *bis*-Amides: Synthesis and In Vitro Evaluation of Novel Promising Agents for Boron Delivery to Tumour Cells

**DOI:** 10.3390/ijms232213726

**Published:** 2022-11-08

**Authors:** Dmitry A. Gruzdev, Angelina A. Telegina, Galina L. Levit, Olga I. Solovieva, Tatiana Ya. Gusel’nikova, Ivan A. Razumov, Victor P. Krasnov, Valery N. Charushin

**Affiliations:** 1Postovsky Institute of Organic Synthesis, Russian Academy of Sciences (Ural Branch), Ekaterinburg 620108, Russia; 2Institute of Cytology and Genetics, Russian Academy of Sciences (Siberian Branch), Novosibirsk 630090, Russia; 3Department of Physics, Novosibirsk State University, Novosibirsk 630090, Russia; 4Nikolaev Institute of Inorganic Chemistry, Russian Academy of Sciences (Siberian Branch), Novosibirsk 630090, Russia; 5Institute of Chemical Engineering, Ural Federal University, Ekaterinburg 620002, Russia

**Keywords:** folic acid, amides, carboranes, cytotoxicity, MTT assay, tumour cells, boron accumulation, BNCT

## Abstract

The design of highly selective low-toxic, low-molecular weight agents for boron delivery to tumour cells is of decisive importance for the development of boron neutron capture therapy (BNCT), a modern efficient combined method for cancer treatment. In this work, we developed a simple method for the preparation of new *closo*- and *nido*-carborane-containing folic acid *bis*-amides containing 18–20 boron atoms per molecule. Folic acid derivatives containing *nido*-carborane residues were characterised by high water solubility, low cytotoxicity, and demonstrated a good ability to deliver boron to tumour cells in in vitro experiments (up to 7.0 µg B/10^6^ cells in the case of U87 MG human glioblastoma cells). The results obtained demonstrate the high potential of folic acid–*nido*-carborane conjugates as boron delivery agents to tumour cells for application in BNCT.

## 1. Introduction

Boron neutron capture therapy (BNCT) is a modern binary approach to tumour treatment. This method involves the combined use of two components: a chemical agent that delivers boron to tumour cells, and irradiation with thermal or epithermal neutrons. Thermal neutrons interact selectively with ^10^B atoms, causing their decay with the emission of high-energy alpha particles and ^7^Li nuclei, and do not have a noticeable effect on cells and tissues consisting of biogenic elements [1,2,3]. Selective delivery of compounds containing one or more ^10^B atoms to tumour cells makes it possible to selectively damage them under the action of thermal neutrons. The main requirements for boron delivery agents suitable for BNCT include: (i) a tumour/healthy tissue distribution index of at least 3:1; (ii) the ability to deliver 20–50 µg ^10^B per 1 g tumour; (iii) minimal toxicity; (iv) high solubility in biological media [4,5,6]. To date, only a few compounds are clinically used for BNCT, namely sodium borocaptate (BSH) [7] and 4-boronophenylalanine (BPA) [8], which were first synthesized in the 1950s–1960s and are characterized by low tumour targeting. A number of more selective boron delivery agents have shown high efficiency in in vivo experiments [9,10,11,12,13,14,15]. The design of many potential agents for BNCT is based on the introduction of boron-containing groups (fragments of boric acid or polyhedral boranes and carboranes) into the structure of natural compounds that can be selectively absorbed by tumour cells [16,17]. In particular, considerable attention is paid to the preparation of carborane-containing derivatives and analogues of natural amino acids and short peptides [18,19,20,21,22,23,24,25].

Folic acid (vitamin B9) is a vital compound for the growth and division of mammalian cells. Cells of various types of tumours are characterized by active expression of folate receptors (FRs) and are able to accumulate folic acid and its derivatives [26,27,28,29]. Modification of xenobiotics [30,31,32,33,34,35], liposomes [36,37,38,39,40], and nanoparticles [41,42,43,44,45,46] using folic acid is a common way to obtain targeted agents for the tumour treatment and imaging.

In recent decades, a number of the folic acid derivatives and analogues containing residues of boric acid [47], dodecaborane [48,49], and dicarba-*closo*-dodecaboranes (*closo*-carboranes) [50] have been proposed as boron delivery agents for BNCT. Such compounds generally require extensive purification, and their preparative yields are not high. Folic acid amide containing a 4-aminophenylboronic acid residue exhibited high hemotoxicity [47]. In vitro experiments have shown that the derivatives of pteroic acid containing dodecaborane residues possess low toxicity and can accumulate in the FR-expressing cells [48,49]. Intratumoural administration (CED, convection-enhanced delivery) made it possible to achieve a significant accumulation of the pteroic acid–decaborane conjugate in F98 glioma cells in vivo [51]. Modification of boron-containing liposomes and nanoparticles by folic acid also made it possible to achieve targeted boron delivery to the tumour [52,53,54,55,56,57,58,59,60,61].

The purpose of our work is to study the possibility of obtaining water-soluble folic acid derivatives containing two carborane residues and to evaluate their toxicity and ability to deliver boron to cells. We tried to synthesize simple-in-structure folic acid derivatives that contained at least 15 wt.% boron and were soluble in water at a concentration of at least 5 mg/mL at a pH close to neutral. We used primary amines containing the fragments of 7,8-dicarba-*nido*-undecaborane (*nido*-carborane) (compounds **1a,b**) and *closo*-carborane (compound **1c**) as boron-containing building blocks. Functionalization of both carboxyl groups of folic acid using compounds **1a–c** makes it possible to obtain diamides containing two carborane residues (18–20 boron atoms in one molecule); the presence of two negatively charged fragments in the molecules of *nido*-carborane derivatives ensures the water solubility.

## 2. Results

### 2.1. Synthesis

*nido*-Carborane-containing amines **1a** and **1b** were synthesized from 3-amino-*closo*-carborane and (*closo*-carboran-1-yl)acetic acid using the coupling reaction with 6-Boc-aminohexanoic acid and Boc-ethylenediamine, respectively, followed by deboronation using caesium fluoride and removal of the protecting groups [62]. 3-(6-Aminohexanoylamino)-*closo*-carborane (**1c**) was obtained as hydrochloride from the corresponding Boc-derivative **2** (Figure 1).

Coupling of amines **1a–c** to folic acid by the carbodiimide method in DMSO in the presence of HOBt and an auxiliary base at a folic acid–amine **1a–c**–EDCI×HCl molar ratio of 1:2.2:2.5 smoothly led to the corresponding *bis*-amides **3a–c** (Figure 2). According to TLC, the reaction was completed in 24 h, and there were no monoamides or unreacted folic acid in the reaction mixture. Previously, a similar approach was used for the synthesis of *bis*-amides of folic acid [63] and methotrexate [64].

Analytical samples of derivatives **3a** and **3b** were obtained by precipitation from the reaction mixture with aqueous HCl followed by washing with acetonitrile; flash-chromatography on silica gel was also used for their purification. *bis*-Amides **3a,b** containing *nido*-carborane fragments were isolated as internal salts (according to elemental analysis data). Their ^1^H NMR spectra contained broad signals in the region of 4.5–8.3 ppm corresponding to protonated secondary and tertiary amino groups in the pteroyl fragment. The characteristic signals of the bridging hydrogen atoms of *nido*-carborane were observed in the region of –3.2…–2.4 ppm, while the signals of the CH groups of the carborane fragment of compounds **3a** and **3b** had a chemical shift of 1.84 and 2.26 ppm, respectively. Compound **3c** containing two *closo*-carborane residues was isolated in pure form after washing the crude reaction product with a 7:3 acetonitrile–water mixture.

While compound **3c** was poorly soluble in water and most organic solvents, *nido*-carborane derivatives **3a** and **3b** were characterized by significant solubility in MeCN and MeOH, and were soluble in water at alkaline pH. Thus, the solubility of conjugates **3a,b** in 0.5% aqueous NaHCO_3_ was 5.0–6.5 mg/mL, which opens up good prospects for biological testing.

### 2.2. Toxicity Assay

The toxicity profile of compounds **3a** and **3b** was studied in the MTT assay [65] on healthy (nontransformed) cells (BJ-5ta human foreskin fibroblasts) and tumour cells (DU 145 human prostate carcinoma, MDA-MB-231 human breast carcinoma, SK-Mel-28 human melanoma, T98G and U87 MG human glioblastomas) (Figure 1). The antitumour agent cisplatin was used as a positive control (at concentrations 10 times lower than those of compounds **3a,b**). Cell viability in negative control samples (without tested compounds in growth medium) was 100 ± 11%.

The folic acid conjugates **3a,b** were moderately toxic to healthy fibroblasts and tumour cells of various lines. At the same time, compound **3a** was slightly more toxic than compound **3b**. Incubation of healthy cells (human foreskin fibroblasts) in the presence of 3-amino-*nido*-carborane derivative **3b** led to a decrease in cell viability by 12% and 29% at concentrations of the test compound of 0.50 and 1.0 mg/mL, respectively (Figure 1a). The (*nido*-carboran-7-yl)acetic acid derivative **3a** exhibited slightly less toxicity against BJ-5ta fibroblasts (decrease in cell viability by 4% and 24% at concentrations of 0.50 and 1.0 mg/mL, respectively). The susceptibility of SK-Mel-28 melanoma cells to the toxic effects of conjugates **3a,b** was comparable to that of healthy cells (cf. Figure 1a,d).

The highest toxicity of the tested compounds was observed against U87 MG glioblastoma cells. The survival of U87 MG cells decreased by 39–61 and 59–69% when incubated in the presence of conjugates **3a,b** at concentrations of 0.25 and 0.50 mg/mL, respectively (Figure 1e).

### 2.3. Evaluation of Boron Accumulation by Cells

Evaluation of boron accumulation by tumour and healthy (nontransformed) cells in in vitro experiments is an essential step on the way to the design of BNCT agents. Candidate compounds must be able to penetrate into tumour cells and be retained in them when administered in a nontoxic dose.

The ability of compounds **3a** and **3b** to deliver boron into cells was tested using the cell lines of BJ-5ta fibroblasts, DU 145 and MDA-MB-231 carcinomas, SK-Mel-28 melanoma, and T98G and U87 MG glioblastomas (Figure 2). In most cases, the tested compounds were used at a concentration of 0.50 mg/mL, which caused the death of no more than 60% of cells in 72 h. In the case of the U87 MG cell line, for which the toxicity of compounds **3a,b** was the highest, the compounds were used at a concentration of 0.25 mg/mL. Incubation was carried out for no more than 8 h to minimize the toxic effect of the compounds.

It has been found that a higher level of boron accumulation by cells is observed during incubation in the presence of 3-amino-*nido*-carborane derivative **3b** compared to (*nido*-carboran-7-yl)acetic acid derivative **3a**. The greatest accumulation of boron was observed in U87 MG glioblastoma cells; moreover, the content of boron in cells during incubation with compound **3b** increased with time (up to 7.0 µg B/10^6^ cells after 8 h) (Figure 2f).

MDA-MB-231 carcinoma cells accumulated up to 2.5 µg B/10^6^ cells when incubated with conjugate **3b** for 1 h; the amount of boron accumulated when compound **3a** was used did not exceed 0.8 µg B/10^6^ cells (Figure 2c). The maximum level of boron accumulation in SK-Mel-28 and T98G cells was the same as in the case of BJ-5ta human foreskin fibroblasts (0.8–0.9 µg B/10^6^ cells) (Figure 2a,d,e). The DU 145 prostate carcinoma cells were characterized by the least accumulation of conjugates **3a,b** (no more than 0.65 µg B/10^6^ cells in the case of conjugate **3a**) (Figure 2b).

## 3. Discussion

New carborane-containing folic acid *bis*-amides **3a–c** were prepared from readily available precursors using easy-to-perform synthetic procedures. Folic acid derivatives **3a,b** containing *nido*-carborane residues and secondary and tertiary nitrogen atoms capable of protonation were isolated as internal salts. The *nido*-carborane derivatives are able to form salts with bases, therefore the presence of two negatively charged *nido*-carborane fragments ensured high solubility of compounds **3a,b** in 0.5–1% NaHCO_3_ aqueous solutions. Thus, these derivatives meet the requirements for potential agents for BNCT, namely high boron content and good solubility in biological media.

Another important requirement for promising agents for BNCT is their low cytotoxicity. Low toxicity at doses sufficient for the accumulation of boron in the tumour guarantees the successful implementation of this method.

The results shown in Figure 1 indicate that cell incubation in the presence of compounds **3a** and **3b** dissolved in 0.5% aqueous NaHCO_3_ practically does not lead to a decrease in cell survival. Compound **3a** containing residues of (*nido*-carboran-7-yl)acetic acid and ethylenediamine was slightly more toxic than derivative **3b** based on 3-amino-*nido*-carborane and 6-aminohexanoic acid.

The low toxicity of folic acid conjugates against SK-Mel-28 melanoma cells may be due to the low level of accumulation of folic acid derivatives in this cell type. Thus, it is known that melanosomal sequestration and cellular export may underlie the resistance of melanoma cells to the action of methotrexate [66] and cisplatin [67].

Literature data on the level of FR expression in glioma cells are rather contradictory. It has been reported that glioma cells, including U87 MG glioblastoma cells, express folate receptors and are able to capture folic acid conjugates [68,69,70,71,72]. But there is also evidence that the U87 MG cells are not very susceptible to the action of folic acid conjugates [73]. In our case, incubation with folic acid *bis*-amides **3a,b** induced the highest toxic effect on the U87 MG cells. This indicates that folic acid derivatives are able to actively penetrate into tumour cells.

It is known that in some cases, folic acid derivatives functionalized at the γ-carboxylic group of glutamic acid have a higher affinity for FRs compared to α-functionalized derivatives [74,75,76]. At the same time, in some cases the ability of α- and γ-derivatives of folic acid to bind to FRs and transport into the tumour is comparable [77,78,79,80]; folic acid diamides also show high selectivity for tumour targeting [81].

Testing of boron accumulation by cells has shown that incubation in the presence of 3-amino-*nido*-carborane derivative **3b** generally provides a higher concentration of boron in cells compared to (*nido*-carboran-7-yl)acetic acid derivative **3a**. Thus, the observed level of boron accumulation in U87 MG glioblastoma cells (up to 7 µg B/10^6^ cells) significantly exceeds the reported results of boron accumulation during incubation with standard BNCT agents (not more than 0.2 µg B/10^6^ cells in the case of BSH, including in the form of targeted liposomes [82,83], and not more than 1.1 µg B/10^6^ cells in the case of BPA [13,48,84]) and with boron-containing analogues of folic acid (at the level of 1.2–1.8 µg B/10^6^ cells for U87 MG glioma and KB carcinoma [47,48]).

The U87 MG cells showed the highest capacity for boron accumulation when incubated with folic acid *bis*-amide **3b**. The MDA-MB-231 carcinoma cells actively expressing FR-α [85,86] were characterised by somewhat lower boron accumulation (up to 2.5 µg B/10^6^ cells). In this case, the amount of boron contained in cells upon incubation with compound **3b** reached a maximum after 1 h and then decreased, which may indicate the presence of mechanisms for the active excretion of folic acid derivatives from cells. The low level of accumulation of conjugates **3a,b** by DU 145, SK-Mel-28, and T98G cells, as well as by fibroblasts, seems to be associated with a significantly lower amount of the surface FRs-α compared to U87 MG and MDA-MB-231 cells. Thus, it is known that DU 145 prostate carcinoma and T98G glioblastoma cells are characterised by low expression of FRs-α [87,88], while SK-Mel-28 cells are capable of cellular export of folic acid analogues.

Apparently, the degree of boron accumulation by cells during incubation with folic acid *bis*-amides **3a** and **3b** correlates with the level of FR-α expression. This indicates that the process of boron accumulation is based on the binding of carborane-containing conjugates **3a,b** to surface FRs-α and subsequent internalization into cells.

The results obtained indicate that folic acid *bis*-amide **3b** containing 18 boron atoms per molecule is suitable for targeted delivery of boron and can be considered as a potential agent for BNCT of FR-α-positive tumours. Moderate cytotoxicity and a high level of accumulation of compound **3b** by glioblastoma cells in in vitro experiments allows us to count on the possibility of using this derivative even with the natural distribution of boron isotopes for the successful implementation of BNCT. Testing the toxicity and biodistribution of compound **3b** and related derivatives in in vivo experiments seems to be a promising direction in the development of new convenient and highly efficient agents for BNCT.

## 4. Materials and Methods

### 4.1. Chemistry General Section

[2-(7,8-Dicarba-*nido*-undecaboran-7-yl)acetylamino]ethylamine (**1a**) and 3-(6-*tert*-butoxycarbonylamino)hexanoylamino-1,2-dicarba-*closo*-dodecaborane (**2**) were obtained according to procedure published elsewhere [62]. Other reagents are commercially available. Solvents were purified according to traditional methods [89] and used freshly distilled.

Melting points were obtained on a SMP3 apparatus (Barloworld Scientific, Staffordshire, UK) and are uncorrected. Optical rotations were measured on a Perkin Elmer 341 polarimeter (Perkin Elmer, Waltham, MA, USA). The ^1^H, ^11^B and ^13^C NMR spectra of compounds **3a,c** and ^1^H NMR spectra of compounds **1c** and **3b** were recorded on a Bruker Avance 500 instrument (Bruker, Karlsruhe, Germany) (500, 160, and 126 MHz, respectively) at ambient temperature. The ^11^B and ^13^C NMR spectra of compounds **1c** and **3b** were recorded on a Bruker DRX-400 instrument (Bruker, Karlsruhe, Germany) (128 and 100 MHz, respectively) at ambient temperature. TMS and BF_3_ × Et_2_O were used as internal and external standards, respectively. NMR spectra of the compounds obtained, see the Appendix A. Microanalyses were carried out using a Perkin Elmer 2400 II automatic analyser (Perkin Elmer, Waltham, MA, USA). Analytical TLC was performed using Sorbfil plates (Imid, Krasnodar, Russia). Flash column chromatography was performed using Silica gel 60 (230–400 mesh) (Alfa Aesar, Heysham, Lancashire, UK). The high-resolution mass spectra were obtained on a Bruker maXis Impact HD mass spectrometer (Bruker, Karlsruhe, Germany), electrospray ionization (ESI) in negative (compounds **3a**, **3b**) or positive mode (for compound **3c**) or atmospheric pressure chemical ionization (APCI) in positive mode (compound **1c**) with direct sample inlet (4 L/min flow rate).

### 4.2. Synthesis

*3-(6-Aminohexanoyl)amino-1,2-dicarba-closo-dodecaborane hydrochloride (**1c**).* Concentrated HCl (4 mL, 47.68 mmol) was added to a cooled (0–5 °C) solution of compound **2** (0.76 g, 2.04 mmol) in 1,4-dioxane (18 mL). The reaction mixture was stirred at room temperature for 2.5 h, then evaporated to dryness under reduced pressure. The residue was dried in vacuo over P_2_O_5_ and KOH at 60 °C. Yield 0.63 g (100%). Colourless hygroscopic powder. ^1^H NMR (500 MHz, DMSO-*d*_6_) δ (ppm): 1.27–1.32 (m, 2H, 2 × H-4 hexanoyl), 1.47–1.57 (m, 4H, 2 × H-3 and 2 × H-5 hexanoyl), 2.20 (t, *J* = 7.4 Hz, 2H, 2 × H-6 hexanoyl), 2.71–2.78 (m, 2H, 2 × H-2 hexanoyl), 1.2–2.8 (br. s, 9H, 9 × BH), 5.08 (s, 2H, 2 × CH carborane), 7.90 (br. s, 3H, NH_3_^+^), 8.30 (s, 1H, NH). ^11^B{H} NMR (160 MHz, DMSO-*d*_6_) δ (ppm): 15.1, 13.5, 10.7, –5.5.^13^CNMR (100 MHz, DMSO-*d*_6_) δ (ppm): 24.77, 25.90, 27.17, 36.80, 39.03, 57.66 (2C), 176.89. HRMS (APCI): *m*/*z* [M+H]^+^ calcd for [C_8_H_25_^11^B_10_N_2_O]^+^: 275.2892, found: 275.2898.

*General Procedure for the Synthesis of Carborane-Containing Folic Acid bis-Amides **3a,b**.* EDCI×HCl (0.37 g, 1.92 mmol) was added to a solution of folic acid dihydrate (0.37 g, 0.77 mmol), amine **1a** or **1b** (1.69 mmol), HOBt hydrate (0.26 g, 1.69 mmol), and NEt_3_ (0.72 mL, 5.16 mmol) in DMSO (17 mL). The reaction mixture was stirred at room temperature for 48 h, then poured into H_2_O (100 mL). 1*N* NaOH (15 mL) was added to the resulting suspension, and the resulting solution was extracted with EtOAc (3 × 25 mL) and *n*-hexane (25 mL). Combined organic layers were washed with 0.5*N* NaOH (20 mL). Aqueous layers were combined and acidified with 4*N* HCl (~8 mL) to pH 1–2 and left at 5–10 °C for 72 h. The precipitate was filtered off, dried, and subjected to flash column chromatography on silica gel (eluent *n*-BuOH–EtOH–16% aq. NH_4_OH 5:7:3). The fractions containing the fast-eluting component were combined and evaporated to dryness under reduced pressure. The residue was washed with 1*N* HCl (15 mL), then dried in vacuo over P_2_O_5_ and KOH. Analytical samples of compounds **3a** and **3b** were obtained by treatment with MeCN (13 mL per 0.5 g of compound) followed by centrifugation (15,000 rpm at 10 °C, 5 min).

*(2S)-2-[(4-{[(2-Amino-4-hydroxypteridin-6-yl)methyl]amino}phenyl)-formamido]-N^1^,N^5^- bis-{2-[(7,8-dicarba-nido-undecaboran-7-yl)acetylamino]ethyl}pentanediamide semihydrate (**3a**).* Yield 0.47 g (69%). Dark orange powder m.p. > 350 °C. [α]^20^−4.5 (578 nm), −5.1 (546 nm) (*c* 0.32, 1*N* NaOH). ^1^H NMR (500MHz, DMSO-*d*_6_) δ (ppm): −2.75 (br. s, 1H, BH), −2.61 (br. s, 1H, BH), −0.45…2.30 (br. m, 18H, 18 × BH), 1.69–1.74 (m, 1H, H-3B Glu), 1.84 (s, 2H, 2 × CH carborane), 1.94–1.99 (m, 1H, H-3A Glu), 2.00–2.05 (m, 2H, 2 × H-2B acetyl), 2.13 (m, 2H, 2 × H-4 Glu), 2.36 (d, *J* = 14.1 Hz, H-2A acetyl), 2.37 (d, *J* = 14.3 Hz, H-2A acetyl), 3.00–3.13 (m, 8H, 4 × CH_2_ ethylenediamine), 4.25 (br. s, 1H, H-2 Glu), 4.60 (s, 2H, CH_2_ pteroyl), 4.75–6.25 (6H, OH, NH^+^, NH_2_^+^ pteroyl and H_2_O), 6.64 (d, *J* = 8.1 Hz, 2H, pteroyl), 7.41 (br. s, 1H, NH), 7.43 (br. s, 1H, NH), 7.67 (d, *J* = 8.1 Hz, 2H, pteroyl), 7.83 (s, 1H, NH), 7.91 (s, 1H, NH), 8.05 (br. s, 1H, NH), 8.18 (br. s, 2H, NH_2_ pteroyl), 8.75 (s, 1H, CH pteroyl). ^11^B{H} NMR (160 MHz, DMSO-*d*_6_) δ (ppm): −37.2, −33.4, −22.2, −17.6, −14.2, −10.7. ^13^C NMR (126 MHz, DMSO-*d*_6_) δ (ppm): 27.31, 31.93, 38.07, 38.13, 38.46, 38.49, 40.41, 42.19, 45.26, 45.31, 45.70, 46.41 (br. s), 53.18, 54.58 (br. s), 111.24 (2C), 121.68, 127.96, 129.02 (2C), 147.54, 148.06, 150.36, 151.96, 152.22, 158.78, 166.08, 171.03, 171.07, 171.81, 171.86. Calcd (%) for C_31_H_57_B_18_N_11_O_6_ × 0.5H_2_O: C 42.15, H 6.62, N 17.44. Found (%): C 42.36, H 6.53, N 17.19. HRMS (ESI): *m*/*z* [M−2H]^2−^ calcd for [C_31_H_55_^11^B_18_N_11_O_6_]^2−^: 437.8011, found: 437.8040; *m*/*z* [M−2H+Na]^−^ calcd for [C_31_H_56_^11^B_18_N_11_NaO_6_]^−^: 899.5993, found: 899.5944.

*(2S)-2-[(4-{[(2-Amino-4-hydroxypteridin-6-yl)methyl]amino}phenyl)-formamido]-N^1^,N^5^- bis-{5-[(7,8-dicarba-nido-undecaboran-3-yl)aminocarbonyl]pentyl}pentanediamide (**3b**).* Yield 0.48 g (67%). Orange powder m.p. 240–245 °C (decomp.) (MeCN). [α]^20^ −38.7 (578 nm), −50.3 (546 nm) (*c* 0.44, 1% NaHCO_3_). ^1^H NMR (500 MHz, DMSO-*d*_6_) δ (ppm): −2.96 (br. s, 2H, 2 × BH), −0.45…2.3 (br. m, 18H, 18 × BH), 1.11–1.19 (m, 4H, 4 × H-4 hexanoyl), 1.28–1.42 (m, 8H, 4 × H-3 and 4 × H-5 hexanoyl), 1.81–1.97 (m, 2H, 2 × H-3 Glu), 1.97–2.00 (m, 4H, 4 × H-2 hexanoyl), 2.06–2.15 (m, 2H, 2 × H-4 Glu), 2.26 (s, 4H, 4 × CH carborane), 2.95–3.03 (m, 4H, 4 × H-6 hexanoyl), 4.24–4.28 (m, 1H, H-2 Glu), 4.59 (s, 2H, CH_2_ pteroyl), 4.8–6.2 (4H, OH, NH^+^, NH_2_^+^ pteroyl), 6.64 (d, *J* = 8.5 Hz, 2H, pteroyl), 6.88 (s, 2H, 2 × NH), 7.65 (d, *J* = 8.5 Hz, 2H, pteroyl), 7.75–7.82 (m, 2H, 2 × NH), 7.97 (br. s), 7.98 (br. s), and 8.02 (br. s) (3H, NH and NH_2_ pteroyl), 8.75 (s, 1H, CH pteroyl). ^11^B{H} NMR (128 MHz, DMSO-*d*_6_) δ (ppm): −38.5, −37.5, −22.4, −21.3, −18.6, −17.6, −12.1, −11.6, −9.9. ^13^C NMR (100 MHz, DMSO-*d*_6_) δ (ppm): 25.58, 25.62, 26.57, 26.67, 28.16, 29.37 (2C), 32.54, 37.33 (2C), 38.95 (2C), 45.09 (br. s, 4C), 46.19, 53.73, 111.78 (2C), 122.27, 128.48, 129.47 (2C), 148.01, 148.05, 150.86, 152.30, 153.01, 159.22, 166.42, 171.99, 172.08, 176.02 (2C). Calcd (%) for C_35_H_65_B_18_N_11_O_6_: C45.18, H 7.04, N 16.56. Found (%): C 44.95, H 6.93, N 16.60. HRMS (ESI): *m*/*z* [M−2H]^2−^ calcd for [C_35_H_63_^11^B_18_N_11_O_6_]^2−^: 465.8324, found: 465.8356.

*(2S)-2-[(4-{[(2-Amino-4-hydroxypteridin-6-yl)methyl]amino}phenyl)-formamido]-N^1^,N^5^- bis-{5-[(1,2-dicarba-closo-dodecaboran-3-yl)aminocarbonyl]pentyl}pentanediamide (**3c**).* EDCI×HCl (0.44 g, 2.28 mmol) was added to a solution of folic acid dihydrate (0.44 g, 0.91 mmol), amine **1c** (0.62 g, 2.01 mmol), HOBt hydrate (0.31 g, 2.01 mmol), NEt_3_ (0.60 mL, 4.29 mmol) in DMSO (16 mL). The reaction mixture was stirred at room temperature for 22 h, then poured into H_2_O (170 mL). The precipitate was filtered off, dried in vacuo, then cold (0–5 °C) 0.15*N* aqueous NaOH (48 mL) was added, and the reaction mixture was stirred at 5 °C for 30 min. The precipitate was separated by centrifugation (12,000 rpm, 15 min), washed with H_2_O (50 mL), and centrifuged again (washing was repeated thrice). The precipitate was dried, treated with a MeCN–H_2_O 7:3 mixture (100 mL) at room temperature, cooled to 5 °C, centrifuged (12,000 rpm, 15 min), washed with a MeCN–H_2_O 7:3 mixture (2 × 40 mL) and dried in vacuo over P_2_O_5_ at 50 °C. Yield 0.44 g (50%). Yellowish powder m.p. 248–253 °C (decomp.). [α]_D_^20^ +2.5 (*c* 0.33, DMSO). ^1^H NMR (500 MHz, DMSO-*d*_6_) δ (ppm): 1.19–1.25 (m, 4H, 4 × H-4 hexanoyl), 1.31–1.40 (m, 4H, 4 × H-3 hexanoyl), 1.43–1.51 (m, 4H, 4 × H-5 hexanoyl), 1.55–2.45 (br. m, 18H, 18 × BH), 1.79–1.87 (m, 1H, H-3B Glu), 1.91–2.02 (m, 2H, H-3A Glu and H-2B hexanoyl), 2.07–2.14 (m, 1H, H-2B hexanoyl), 2.15–2.17 (m, 4H, 2 × H-4 Glu and 2 × H-2A hexanoyl), 2.97–3.04 (m, 4H, 4 × H-6 hexanoyl), 4.25–4.29 (m, 1H, H-2 Glu), 4.49 (d, *J* = 5.8 Hz, 2H, CH_2_ pteroyl), 5.07 (s, 4H, 4 × CH carborane), 6.63 (d, *J* = 8.6 Hz, 2H, pteroyl), 6.81 (br. s, 1H, NH), 6.92–6.94 (m, 2H, NH_2_ pteroyl), 7.65 (d, *J* = 8.6 Hz, 2H, pteroyl), 7.77 (t, *J* = 5.3 Hz, 1H, NH), 7.79 (t, *J* = 5.5 Hz, 1H, NH), 8.22 (s, 2H, 2 × NH aminocarborane), 8.64 (s, 1H, CH pteroyl), 11.41 (s, OH). ^11^B{H} NMR (160 MHz, DMSO-*d*_6_) δ (ppm): –15.1, –13.5, –10.7, –5.6. ^13^C NMR (126 MHz, DMSO-*d*_6_) δ (ppm): 24.46, 24.49, 25.85, 25.95, 27.68, 28.81 (2C), 32.06, 36.48 (2C), 38.31 (2C), 45.88, 53.17, 57.01 (4C), 111.11 (2C), 121.46, 127.86, 128.91 (2C), 148.43, 148.60, 150.67, 153.63, 156.51, 160.73, 166.02, 171.48, 171.51, 176.47 (2C). Calcd (%) for C_35_H_63_B_20_N_11_O_6_: C 44.24, H 6.68, N 16.22. Found (%): C44.26, H 6.85, N 16.20. HRMS (ESI): *m*/*z* [M+Na]^+^ calcd for [C_35_H_63_^11^B_20_N_11_NaO_6_]^+^: 976.6790, found: 976.6796.

### 4.3. Cell Lines

The following cell lines were used: BJ-5ta human foreskin fibroblasts (ATCCCRL-4001™), U87 MG human glioblastoma (ATCCHTB-14™), T98G human glioblastoma (ATCCCRL-1690™), SK-Mel-28 human melanoma (ATCCHTB-72™), MDA-MB-231 breast carcinoma (ATCCCRM-HTB-26™), and DU 145 prostate adenocarcinoma (ATCCHTB-81™), stored in the SPF-vivarium cryobank at the Institute of Cytology and Genetics of the Russian Academy of Sciences (Siberian Branch), Novosibirsk. Cells were cultured in 5% CO_2_ in DMEM/F12 (1:1) nutrient medium (Biolot, St. Petersburg, Russia) supplemented with 10% fetal bovine serum (Invitrogen, Waltham, MA, USA). Cells were counted on a Countess automatic cell counter (Invitrogen, Waltham, MA, USA).

### 4.4. MTT Cytotoxicity Assay

Cells were seeded in 96-well plates in the amount of 2 × 10^4^ cells per well and cultivated for 24 h. Stock solutions of compounds **3a** and **3b** in 0.5% aqueous NaHCO_3_ (concentration 5.0 mg/mL) were prepared under stirring for 10 min, then incubated at 37 °C for 20 min and sonicated on a Sonicator Q700 ultrasonic homogenizer (Qsonica L.C.C, Newtown, CT, USA) for 30 min. A solution of the commercial anticancer agent Cisplatin (Cisplatin Teva, Pharmachemie B.V., Haarlem, the Netherlands) with an initial concentration of 0.50 mg/mL was used as a positive control. Nutrient medium without additives was used as a negative control. The stock solutions of compounds **3a,b** were added to the nutrient medium with cells in a volume of 1/5 of the total volume of the medium in the well, as a result of which the concentration of compounds **3a** and **3b** in the medium was 1.00 mg/mL; cisplatin concentration was 0.10 mg/mL. Then, a series of twofold dilutions of stock solutions was prepared and added to the nutrient medium with cells in such a way as to obtain nutrient media with the concentration of compounds **3a** and **3b** of 0.50, 0.25, 0.125, 0.063, 0.031, 0.016, and 0.008 mg/mL. The concentration of cisplatin in the positive control samples was 0.10000, 0.0500, 0.0250, 0.0125, 0.0063, 0.0031, 0.0016, and 0.0008 mg/mL. The duration of cell incubation was 3 days at 37 °C in an atmosphere containing 5% CO_2_. After that, the culture medium was removed from each well, a solution of MTT [3-(4,5-dimethylthiazol-2-yl)-2,5-diphenyltetrazolium bromide] in DMEM/F12 (1:1) culture medium (MTT concentration 5 mg/mL) was added and incubated for 4 h; then, the supernatant was removed, and the formazan precipitate was dissolved in DMSO (100 µL). The optical density of the resulting solutions was determined on a Multiskan Sky High Microplate Spectrophotometer (Thermo Fisher Scientific, Waltham, MA, USA) at a wavelength of 595 nm. Cell viability was determined based on optical density; cell viability in the negative control was taken as 100%. Experiments were performed in three parallel runs (for detailed information on cell viability of various cell lines, see the Appendix A).

### 4.5. Boron Uptake and Accumulation Assay

Stock solutions of compounds **3a** and **3b** were prepared in 0.5% aqueous NaHCO_3_ (concentration 5.0 mg/mL). BJ-5ta, SK-Mel-28, T98G, DU 145, MDA-MB-231, and U87 MG cells were cultured in 5 mL of nutrient medium at 37 °C in a 5% CO_2_ atmosphere until a monolayer was obtained (from 3 × 10^6^ up to 5 × 10^6^ cells). The nutrient medium was removed, a mixture of the nutrient medium (4.75 mL in the case of U87 MG cells or 4.50 mL in other cases) and the stock solution of compound **3a** or **3b** (0.25 mL in the case of U87 MG cells or 0.50 mL in other cases) was added to the cells and incubated at 37 °C in a 5% CO_2_ atmosphere. Cells cultured without the addition of test compounds were used as controls.

After the cells were incubated for various times (10 min, 30 min, 1 h, 3 h, 6 h, and 8 h), the culture medium was separated from the cells, and the cells were removed from the substrate with a trypsin–versene solution (1:1) (Biolot, St. Petersburg, Russia), and the number of cells was counted (for the number of cells used for assay, see the Appendix A). The resulting cell suspension was divided into three equal parts, centrifuged (1000 rpm, 5 min), and the cells were separated from the supernatant. 16*M* Nitric acid (1.0 mL) was added to the resulting cells, the mixture was kept at 95 ± 1 °C for 30–40 min, then cooled to 20 °C, and deionized water (3.0 mL) was added. The boron content in the obtained solutions was determined on an iCAP 6500 DUO high-resolution atomic emission spectrometer with inductively coupled plasma (Thermo Fisher Scientific, Waltham, MA, USA) according to the procedure described in [90].

## 5. Patents

Gruzdev, D.A.; Krasnov, V.P.; Telegina, A.A.; Levit, G.L.; Solovieva, O.I.; Razumov, I.A.; Kanygin, V.V.; Gusel’nikova, T.Ya.; Charushin, V.N. *nido*-Carborane-containing folic acid *bis*-amides for boron delivery to tumour cells, Pat. Appl. RU2022123758A (priority 2022-09-07).

## Data Availability

Not applicable.

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
