# Peer review of "Carborane-Containing Folic Acid bis-Amides: Synthesis and In Vitro Evaluation of Novel Promising Agents for Boron Delivery to Tumour Cells"

_ijms, 2022, doi:10.3390/ijms232213726_

Round 1
Reviewer 1 Report
This manuscript describes synthesis of two carborane-attached folates for BNCT, their cytotoxicity and boron accumulation to cells. There have already been a few reports concerning carborane-attached folate as the author is citing (ref. 48-50). However, this manuscript involves more detailed cytotoxicity data and boron accumulation data. Correlation of the boron accumulation with FR-a expression indicates the folate part successfully works.
I have only one question as bellow:
In Scheme 1, the author used CsF on the synthesis of 1b. However, I cannot understand the role of CsF for the synthesis. The author should explain the role.
Description of Figure 2 should be improved. It should be better to understand if the name of the cell is described in each graph.
Author Response
Dear Reviewer,
The authors are greatful for valuable comments and remarks. We believe that our responses to these comments and revisions we made will improve our manuscript and make it suitable for publication in the International Journal of Molecular Sciences.
Comments and Suggestions for Authors
In Scheme 1, the author used CsF on the synthesis of 1b. However, I cannot understand the role of CsF for the synthesis. The author should explain the role.
Response: Cesium fluoride (CsF) is a convenient reagent for the conversion of closo-carborane derivatives to the corresponding nido-derivatives (see, for example, [Yoo, J. et al. Inorg. Chem.2001, 40, 568-570; Wei, X. et al. Organometallics 2006, 25, 609-621; Gruzdev, D.A. et al. J. Org. Chem. 2022, 87, 5437-5441]). Fluoride ion provides detachment of the BH vertex closest to the CH vertices of a closo-carborane. In the case of carboranyl-acetic acid derivatives, the B(3)H and B(6)H vertices are equivalent, and deboronation leads to compound 1a as a mixture of enantiomers. In the case of compound 1b, treatment of amide 2 with cesium fluoride in ethanol leads to the selective removal of the B(6)H vertex and does not affect the B(3)N vertex. We have made some addition to the text of manuscript, see line 85 in pdf file.
Description of Figure 2 should be improved. It should be better to understand if the name of the cell is described in each graph.
Response: Figures 1 and 2 have been improved.
On behalf of all coauthors.
Sincerely yours,
Dr. Dmitry A. Gruzdev
Reviewer 2 Report
The authors report the synthesis and characterization of carborane-containing folate amide derivative by cytotoxcity assay against normal fibroblast and a panel of cancer cell lines.
The overall study is technically sound and fits the scope of IJMS. The work has potential positive impact in the cancer therapy field.
One minor question, is it possible to include HEK or CHO type cell lines as a control "normal" cell line as well since that's a common cell line many researchers use. Even if preliminary data on toxicity and/or update will be very helpful to understand the extent of broad tolerance by "normal" cell lines.
Author Response
Dear Reviewer,
The authors are greatful for valuable comments and remarks.
Comments and Suggestions for Authors:
One minor question, is it possible to include HEK or CHO type cell lines as a control "normal" cell line as well since that's a common cell line many researchers use. Even if preliminary data on toxicity and/or update will be very helpful to understand the extent of broad tolerance by "normal" cell lines.
Response: It should be noted that HEK-293 and CHO cells differ from normal human cells in the number of chromosomes (they are hypotriploid and hypodiploid, respectively). In our study, we have chosen BJ-5ta cell culture (diploid human cell line of male origin with a modal chromosome number of 46 that occurred in 90% of the cells counted) as a control; these cells are closest to the ‘normal’ nontransformed human cells. We are currently continuing biological testing of the lead compound (3b), including toxicity assessment on other cell types.
On behalf of all coauthors.
Sincerely yours,
Dr. Dmitry A. Gruzdev